# The Effects of Irrigation, Genotype and Additives on Tef Silage Making

**DOI:** 10.3390/ani13030470

**Published:** 2023-01-29

**Authors:** Philip Wagali, Chris Sabastian, Yehoshua Saranga, Shiran Ben-Zeev, Sameer J. Mabjeesh

**Affiliations:** 1Department of Animal Sciences, The Robert H. Smith Faculty of Agriculture, Food, and Environment, The Hebrew University of Jerusalem, Rehovot 7610001, Israel; 2The Robert H. Smith Faculty of Agriculture, Food, and Environment, The Robert. H. Smith Institute of Plant Sciences & Genetics in Agriculture, The Hebrew University of Jerusalem, Rehovot 7610001, Israel

**Keywords:** tef, silage, digestibility, additives

## Abstract

**Simple Summary:**

Tef is a multi-harvest crop with high production capacity and outstanding fodder quality. Our overall goal is to establish tef as a multi-harvest summer fodder crop. Here, we report on the effects of irrigation (75% vs. 100%), genotype (RTC-2, RTC-119, RTC-361, and RTC-400), and additives; heterofermentative inoculum, molasses, and molasses + heterofermentative inoculum on tef silage making and silage quality. Our results showed for the first time that tef can be ensiled. Most of tef silage qualitative parameters were better at 75% irrigation than 100%, and additives treatments improved the silage parameters; additives lowered silage pH and ammonia nitrogen, but increased in vitro dry matter digestibility, lactic acid, lactic acid bacteria, and crude protein content of tef silage.

**Abstract:**

Tef is known as a multi-harvest crop with high production capacity and outstanding fodder quality. Hence, our overall goal is to develop tef as a new multi-harvest summer crop that can maintain high-quality feed and contribute to both field crops and the livestock industry in Israel. In this study, we aimed to evaluate the ability to preserve tef as silage. Four tef genotypes grown under well-watered (100%) and water-limited (75%) irrigation regimes were harvested at grain filling stage and ensiled with either no additives (control, CTL), or with heterofermentative inoculum (HI), molasses (MOL), and both MOL + HI. Our results showed for the first time that tef could be ensiled, although water-soluble carbohydrates (WSC) were lower than those in corn, “the perfect ensiling crop”. Most of the tef silage qualitative parameters were better at water-limited irrigation. Additives HI or MOL or MOL + HI also improved silage parameters, e.g., lowered pH and ammonia nitrogen content, but increased in vitro dry matter digestibility, lactic acid and crude protein content, and lactic acid bacteria counts of tef silage. The current results imply increasing the diversity of local ruminant fodder crops, ensuring high-quality feed supply during the summer.

## 1. Introduction

Tef (*Eragrotis tef* (Zucc.) Trotter, also known as teff) is a cereal plant of Ethiopian origin, known as an agricultural crop for over 2000 years [1]. Tef is a major source of food and feed in Ethiopia. It is a basic product providing a significant portion of the daily diet (bread) of the population and animal feed, occupying about 30% of the land used to cultivate grains in the country [2]. Tef is an annual cereal plant with an upright stem; inflorescences are oat-like panicles which may be free or compact and comprise 2–12 flowers per spikelet [3]. The flowers are bisexual with about 99% self-pollination. Tef has very small grains and 1000 seeds have a mass ranging from 0.25 to 0.35 g with varieties differing in grain colour from white to dark brown. Tef has a C4 photosynthetic pathway, which enables high productivity under high radiation and temperature. Tef can grow in a wide variety of environments (marginal lands inclusive) [4,5,6]. The plant can adapt to low rainfall conditions (450–550 mm), and it can tolerate moisture stress [3]; however, under severe drought there can be a great yield reduction (over 70%) [2]. On the other hand, excess irrigation leads to plant lodging, causing loss of yields, and hampers mechanical harvest [7,8,9,10].

In recent years, many countries outside Ethiopia (Israel-inclusive) have increased interest in tef cultivation as a grain or fodder crop [11,12]. In Israel, the interest in tef is derived from various reasons. There is a demand of approximately 8000 tonnes of grains per annum by the Ethiopian ethnic community, while the Ethiopian government restricts the export of tef. There is also potential in domestic and export “health food” markets for tef kernels and products [9]. Lastly, there is a great interest to diversify the field crops in general as well as the ruminant fodder crops in Israel. Dairy cows’ feeds in Israel are based on locally produced roughage (fibrous material), mostly wheat and maize, complemented with expensive imported concentrate (rich in energy and digestible nutrients). The introduction of tef as a new, high quality fodder crop will therefore contribute to the diversity of the forage crops.

Tef straw is preferred by livestock relative to other cereal straw, and hence could be used as livestock feed during the dry seasons when there is a shortage of feed, fetching a premium price [3,13]. Research has shown that the nutritional value of tef fodder is like other fodder used to make hay and silages [14,15] with a digestibility value similar to tropical grasses [16]. Tef has a higher or similar crude protein (CP) content compared to other cereal grains [17,18] as well as high trace minerals such as calcium, phosphorus, iron, etc., and thiamine [19]. Hence, it can be used as an alternative summer forage [17].

Silage is one of the most reliable methods to preserve feedstuffs. Over the years, there has been a better understanding of the biochemistry and microbiology of the ensiling process paving way for development of numerous silage additives [20,21]. High nutritive value of silage depends on harvesting the crop at the proper stage of maturity, activity of plant enzymes, epiphytic microorganisms, and encouraging dominance of lactic acid bacteria (LAB) [22]. Research has shown that the various additives had positive effects on silages as well as animal performance, although this greatly varied [23,24].

As part of a project that aims to assess tef as a multi-harvest summer crop for Israeli dairy cows, the objective of this study was to evaluate the ability to preserve tef as silage as well as assess the effects of cultivars, irrigation regime, and additives on tef silage.

## 2. Materials and Methods

### 2.1. Plant Materials and Growth Conditions

A panel representing the wide phenotypic diversity available in tef was assembled at the Robert H. Smith Faculty of Agriculture, Food, and Environment, The Hebrew University of Jerusalem and tested across several years [25]. Four tef lines (RTC-2, RTC-119, RTC-361, and RTC-400) with favourable forage characteristics (high productivity with high proportion of leaf biomass) and similar, medium-early heading time (~57 days after sowing) were selected for this experiment.

All four genotypes were sown on 16 April 2020 at Eden Experimental Farm, Beit She’an Valley, Israel (32°28′20″ N, 35°29′10″ E). The experimental field was divided into two main plots (for the two irrigation regimes, 75% and 100% irrigation), each consisting of four semi-commercial sub-plots measuring 185 × 10 m. Border plots were sown between two main plots as well as on both sides of the field as shown in (Appendix A) in Supplementary File. Sprinkler lines were placed between each two plots in the centre of a 2-m unsown strip. The experimental site is characterized by dry hot summer; average minimum and maximum temperatures during the experimental season were 18.0 °C and 33.0 °C, respectively. During the first couple of weeks, small amounts of irrigation were applied every other day (total 145 mm), uniformly across the entire experimental plot, to ensure plant establishment, followed by twice weekly differential irrigation. The four tef genotypes were grown under two irrigation treatments, well-watered (100%) and water-limited (75%), which received total amounts of 436 mm and 327 mm, respectively (including the early uniform irrigations). Solid Urea fertilizer was applied at a rate of 53 kg/ha on 18 May 2020. Agronomic practices were based on the recommendations for setaria (*Setaria italica*) cultivation, since tef had no recommended management scheme for that region.

Tef plant materials for the silage-making experiment were collected following mechanical harvest from all four genotypes under both irrigation regimes at grain-filling stage (10–14 days after heading) on 21 June 2020, 67 days after sowing.

### 2.2. Silage Making

The plant materials for ensiling were chopped to a size of approximately 6 cm, and each sample divided into four additives treatments to encourage proper fermentation: control (CTL), 0.1% (1 mL/kg of fresh ensiling material), liquid heterofermentative inoculum (HI; EM-Zoo^®^; Aseret, Israel), 0.5% (5mL/kg of fresh ensiling material), a solution of sugarcane molasses (MOL), which can be categorised as a stimulant or source of substrate in the fermentation process [20], and both additives combined (MOL + HI). The use of HI, also known as effective microorganisms (EM), is a technology developed by Professor Teruo Higa at the University of Ryukyus, Okinawa, Japan in 1982 [26,27]. The liquid microbial inoculum preparation contains beneficial bacteria (including LAB) in the form of yoghurt, grown on a medium of sugar cane molasses and water [28]. The HI operates on the principle of competitive exclusion [29,30]. Bacteria produced in the inoculum secrete materials that support life, health, and rejuvenation, including vitamins, enzymes, antioxidants, amino acids, and more according to [27,31]. By doing so, they create in their environment a positive and powerful presence that makes it difficult for pathogenic microorganisms to reside and thrive [31]. The control served to assess the possibility of ensiling tef, whereas the other three treatments aimed to assess the effect of additives on tef silage. The additives (HI, MOL, or MOL + HI) were dissolved in 50 mL of distilled water and mixed with the ensiling materials. The same amount (50 mL) of distilled water was added to the control. Samples of approximately 300 g were vacuum-sealed in marked polythene bags with a known weight. For each combination of genotype × treatment, there were four or two replicates of the ensiling material per treatment (total 8 or 16 samples per genotype per treatment). The vacuum-sealed silage samples were kept at room temperature for 90 days, the bags were weighed before opening, and chemical analyses and in vitro digestibility assays were done on the silage as specified below.

### 2.3. Chemical Analyses of Ensiling Material

In order to characterise the raw material, dried samples of the four genotypes (original material prior to ensiling) were first ground in a hammer mill with a 4mm screen and later to finer particles in a knife mill (Thomas–Willey Laboratory Mill, model 4, Arthur H. Thomas Company, Philadelphia, PA, USA) to pass through a 2 mm screen. The samples were analysed for dry matter (DM), Organic Matter (OM), and Crude Protein (CP) using the Kjeldahl method (KjelMaster-K375, Buchi, Switzerland) [32], Neutral Detergent Fiber (NDF), Acid Detergent Fiber (ADF), hemicellulose, cellulose, Acid Detergent Lignin (ADL) [33], and Water Soluble Carbohydrates (WSC) [34], and in vitro digestibility as described hereafter.

### 2.4. Chemical Analyses of Tef Silage

Each of the opened vacuum silage bags after 90 days (day 0), 100 g sample of silage was agitated with 400 mL of distilled water in special nylon bags using a shaker (STO-4 Stomacher shaker MRC Ltd., Holon, Israel) set at a paddle speed of 10/s for 3 min. The filtrate was used for pH measurement using a pH meter (Sartorius AG, Gottingen, Germany). A portion of the filtrate was used for microbial analysis; enumeration of LAB was performed using Rogosa SL Agar (Himedia^®^, Mumbai, India) according to [35], and enumeration of moulds and yeast was performed using Malt Extract Agar (Sigma_Aldrich^®^, Rehovot, Israel). Other portions of the filtrate were stored at −20 °C and later were used to measure volatile fatty acids (VFA) analysis [36] using *hp* gas chromatograph (model 5890A), lactic acid (LA) concentration using a calorimetric method [37], WSC content using a calorimetric method [34], and ammonia nitrogen concentration (N-NH_3_) using a calorimetric method [38]. LA, WSC, and N-NH_3_ were read in 96-well plates (F96 MaxisorpNunc-Immuno plates, Rosklide, Denmark) in a spectrophotometer (Biotek ELx808, Lumitron Ltd.). Another silage sample was dried at 60 °C for 48 h and analysed for DM and OM [32] as well as NDF, ADF, hemicellulose, cellulose, and ADL [33]. This sample was used for in vitro dry matter digestibility (IVDMD) according to the protocol of [39] adapted to DAISY^II220^ apparatus (ANKOM Technology Corp., Fairport, NY, USA). The IVDMD was completed in two phases: anaerobic and aerobic of 48 h incubation each. The in vitro dry matter digestibility for 240 h (IVDMD240) was aimed to measure the digestibility of NDF after 240 h of incubation. Rumen fluids were obtained from two ruminally fistulated wether Assaf sheep which were fed a standard ration containing 2.42 Mcal ME, 120 g of CP per kg DM basis. Ration contained 73% roughage feeds of mainly wheat silage, clover hay, wheat hay, and the rest concentrates, minerals, and vitamins to satisfy the maintenance requirements according to the recommendations of [40]. All procedures involving animal experiments were approved by the IACUC of the Hebrew University of Jerusalem (ethical approval number AG-14102).

### 2.5. Aerobic Stability Assay

A sample (150 g) was incubated for 5 days at room temperature as described by [41]. At the end of the incubation, 100 g of sample was blended with 400 mL of distilled water. The resultant filtrate was used for pH measurement and microbial analysis (moulds and yeast). The rest of the filtrate was stored at −20 °C and later analysed for LA, VFA, WSC, and N-NH_3_. A sample of silage from the aerobic stability experiment was also dried at 60 °C for 48 h, ground through a 2mm mesh, and analysed for DM, OM NDF, ADF, hemicellulose, cellulose, and ADL and assayed for IVDMD.

### 2.6. Statistical Analysis

Data was exported from an excel sheet to JMP^®^ Pro (VERSION 16.0.0, SAS Institute Inc., Cary, NC, USA) and subjected to a two-way ANOVA (irrigation × genotype) for ensiling materials or three-way ANOVA (irrigation x genotype x additive) for silage quality (pH, LAB, LA, WSC, N-NH_3_, and VFA) and chemical composition (OM, CP, NDF, ADF, hemicellulose, cellulose, ADL contents) as well as IVDMD and DM loss. The data was further subjected to Dunnett test to compare the control to the additive treatments at *p* < 0.05. To compare the effect of irrigation, genotype, and additives on tef silage, a Tukey HSD test was performed at *p* < 0.05.

## 3. Results

### 3.1. Chemical Composition and Digestibility of Tef Ensiling Material

Chemical compositions of the tef ensiling material (biomass) were significantly affected, in most cases, by genotypes, irrigation regime, and their interaction, whereas the in vitro digestibility variables did not show such trend (Table 1 and Table 2). There was a significant interaction between irrigation and genotype for DM content, where in 75% irrigation, RTC-119 showed higher DM than RTC-400 and were intermediate for RTC-2 and RTC-361. Meanwhile, in 100% irrigation, RTC-119 showed the highest DM, followed by RTC-400 and RTC-361 with RTC-2 having the lowest value. An interaction effect excited on OM content, where in 75% irrigation treatment, RTC-361 and RTC-400 had the higher value of 93.0%, and RTC-2 and RTC-119 showed the lowest value of 90.6%. Meanwhile, at 100% irrigation, OM contents were highest for RTC-119 and RTC-361 (90.7%), and the lowest values were showed in RTC-2 and RTC-400 (88.2%). The CP content also had a significant interaction effect as shown in Table 1. At 75% irrigation regimen, genotypes RTC-361 had the highest value (14.8%) followed by RTC-2 (14%) and lowest for RTC-119 (12.2%), and RTC-400 was similar to RTC-361 and RTC-2. Meanwhile, at the 100% irrigation, RTC-2 had the highest value (15.1%), followed by RTC-119 and RTC-400 (14.6%), and RTC-361 showed the lowest value (13.1%). A similar interaction effect existed for NDF content where at 75% irrigation, RTC-2 and RTC-119 had higher values than RTC-361 and RTC-400. However, at 100% irrigation NDF content was highest for RTC-119 followed by RTC-2 and RTC-361 while RTC-400 was similar to the other genotypes. A significant interaction effect of irrigation and genotype existed on ADF content; at 75% irrigation treatment, RTC-2 and RTC-119 had the highest value (32.4%) while RTC-361 and RTC-400 had the lowest content (27.4%). Looking at the 100% irrigation, RTC-119 had the highest value of 35.1% followed by RTC-2 and RTC-361, and RTC-400 was similar to the rest of genotypes. No significant interaction affect between irrigation and genotypes was found for hemicellulose contents, nor was a significant effect of genotype observed; however, a significant irrigation effect existed where 75% irrigation presented lower hemicellulose content (34.8%) than 100% irrigation (36.2%). Looking at ADL content, a significant interaction effect was present between irrigation regimen and genotypes; at 100% irrigation, RTC-119 had the higher value and lowest for RTC-2, while RTC-361 and RTC-400 were intermediate to RTC-119 and RTC-2, respectively. At 75% irrigation treatment, all genotypes showed the same value averaging 3.67%. It is worth noting that WSC were less than 1% in all irrigation treatments and genotypes. In vitro NDF digestibility (IVNDFD) measured after 48 h of incubation showed a significant interaction effect between irrigation treatment and genotypes (Table 2). At 75% irrigation, the highest value (54.7%) was found for RTC-2 followed by the rest of the genotypes (45.3%) while, at 100% irrigation, RTC-400 was found to be the highest (53.7%) and RTC-119 and RTC-361 intermediate (45.8%), and the lowest value (42.4%) was showed for RTC-2. However, these differences were ≤ 3% units. The IVNDFD after 240 h incubation (IVNDFD240) was higher on average by 32% when compared to 48 h under both irrigation treatments (62.1% vs. 46.8%, respectively). 

In order to reduce the volume of data presented hereafter and in accordance with our main objectives, the data presented hereafter are averaged across genotypes, hence focusing on the effects of additives and irrigation regimes. Complete data set for irrigation, genotype, and additive can be found in the Supplementary Material (Appendix A).

### 3.2. pH of Tef Silage

Results of tef silage at day 0 showed that pH was lower at 75% than 100% irrigation regime (Table 3). Furthermore, results also showed that pH was statistically different among the additives treatments and was lowest in MOL + HI and highest in CTL (4.4, 4.5, 4.6, and 4.6 for MOL + HI, MOL, HI, and CTL, respectively, averaged across irrigation). Similar trends were observed for pH at 5 days after aerobic exposure for both the effects of irrigation and additives.

### 3.3. In Vitro Dry Matter Digestibility of Tef Silage

IVDMD at day 0 was higher at 75% irrigation than 100% irrigation, implying 75% was better than 100% (Table 3). IVDMD results also showed significant differences among additives with the lowest IVDMD in CTL and highest in MOL + HI (*p* < 0.0231; 59.6, 59.6, 61.8, and 62.7% for CTL, HI, MOL, and MOL + HI, respectively, averaged across irrigation), where we could infer that additives improve IVDMD of tef silage. Results of IVDMD after aerobic exposure for 5 days had a similar trend to day 0, but with slightly reduced IVDMD for irrigation effect. IVDMD at day 5 was significantly different, lowest in EM and highest in CTL (*p* < 0.0323; 57.2, 57.0, 55.4, and 53.4% for CTL, MOL + HI, MOL, and HI, respectively, averaged across irrigation).

### 3.4. Microbial Properties of Tef Silage

Colony counter (expressed as log10 CFU g/DM) results showed that LAB was lower in 75% than 100% irrigation (Table 3). LAB was different among additives with MOL having the lowest and HI the highest (*p* < 0.0109, 8.9, 9.5, 10, and 10.1 log10 CFU g/DM for MOL, MOL + HI, CTL, and HI, respectively, averaged across irrigation). It is worth noting that tef silage did not grow moulds on both day 0 and 5 days after aerobic exposure (Appendix A).

### 3.5. VFA of Tef Silage

At day 0, results showed that additives increased acetic acid concentration; it was lowest in CTL and highest in HI (*p* < 0.0488; 0.51, 0.69, 0.87, and 0.90 g/100 g DM for CTL, MOL, MOL + HI, and HI, respectively, averaged across irrigation) (Table 4). Total VFA was lowest in CTL and highest in MOL + HI and significantly different (*p* < 0.0229; 0.54, 0.76, 0.94, and 1.06 for CTL, MOL, HI, and MOL + HI, respectively, averaged across irrigation). After 5 days of aerobic exposure, irrigation increased acetic acid concentration in tef silage. It tended to be significantly different (*p* < 0.0517), lower in 75% than 100% irrigation. Propionic acid concentration was 0.16 g/100 DM in 75% and was not detected in 100% irrigation (*p* < 0.0006). Additives decreased propionic acid, and it was 0.12 g/100 DM in MOL and 0.20 g/100 DM in CTL. It was not detected in HI and MOL + HI (*p* < 0.0033). It is worth noting that ethanol and butyric acid were negligible and did not differ between irrigation and additives.

### 3.6. Lactic Acid, Water Soluble Carbohydrates and Ammonia Nitrogen of Tef Silage

Results of lactic acid concentration, following 5 days of aerobic exposure, showed that 75% irrigated tef was more than two-fold higher than 100% irrigation (Table 5). Our results further showed that lactic acid was significantly different among additives treatments, with the highest in MOL (more than double) and the lowest in HI (*p* < 0.0058; 0.9, 1.0, 1.2, and 2.0 g/100g DM for HI, MOL + HI, CTL, and MOL, respectively, averaged across irrigation).

Water-soluble carbohydrates content on the day of opening the silage were higher at 75% irrigation than 100% irrigation (Table 5). Additives decreased WSCs; they were four times higher in CTL than HI, which was the lowest (*p* < 0.0001; 0.04, 0.06, 0.08, and 0.16 g/100 g DM for HI, MOL + HI, MOL, MOL, and CTL, respectively, averaged across irrigation). After 5 days of aerobic exposure, WSCs showed a trend similar to that of day 0 (*p* < 0.0001; 0.10, 0.13, 0.06, and 0.22 g/100g DM for MOL + HI, MOL, HI, and CTL, respectively)**.**

Ammonia nitrogen concentration was lower at 75% irrigation than 100% irrigation (Table 5). Additives reduced N-NH_3_ concentration, and a comparison among additives treatments showed that N-NH_3_ tended to be significantly different, was lowest in MOL + HI, and highest in CTL (*p* < 0.0608; 3.0mg/dL for, MOL + HI, 4.7 mg/dL for both MOL and HI, and 5.0mg/dL for CTL, averaged across irrigation). After aerobic exposure for 5 days, the trend for N-NH_3_ was similar to day 0 for irrigation with the concentration of more than double in 100% than 75% irrigation.

### 3.7. Chemical Composition of Tef Silage

Our results showed that OM content at day 0 was significantly higher at 75% irrigation than 100% irrigation (Table 6). For the additives treatments, OM content was lowest in MOL and highest in CTL (*p* < 0.0029; 89.4, 89.7, 89.9, and 90.2% for MOL, MOL + HI, HI, and CTL, respectively, averaged across irrigation). Day 5 after aerobic exposure showed the same trend but fell below the significance threshold.

Crude protein content at day 0 was higher at 75% than 100% irrigation (Table 6). The CP content was the highest in MOL + HI and lowest in MOL (*p* < 0.0315; 13.7 for MOL, 14.0 for both HI and CTL, and 14.4% for MOL + HI, respectively, averaged across irrigation). The trend for CP at day 5 after aerobic exposure was similar to that of day 0; however, CP content was lowest in CTL and highest in MOL (*p* < 0.0181; 13.5% for CTL, 14.2% for both MOL + HI and HI, and 14.3% for MOL).

Unlike CP, NDF content at day 0 showed that in 75% irrigation regime was less than 100% by ≈4% units (Table 6). After 5 days of aerobic exposure, the NDF content followed a similar trend for irrigation with ≈2% units greater in 100% than 75% irrigation.

The trend for ADF content was identical to that of NDF content for both days 0 and day5, but with higher ADF values for day 5 after aerobic exposure (Table 7). Irrigation effect was significantly different with approximately 2 and 4 more % units in 100% irrigation than 75% irrigation for day 0 and 5, respectively.

Hemicellulose content exhibited a trend similar to NDF and ADF content, except with just one unit lower at 75% irrigation than 100% irrigation (Table 7). Hemicellulose was significantly different among additives treatments. It was lowest in MOL + HI and highest in HI (*p* < 0.0093; 29.9, 29.5, 28.9, and 28.8% for MOL + HI, MOL, CTL, and HI, respectively, averaged across irrigation). Day 5 results showed that 100% was less than 75% irrigation by 2 units and additives treatments, and MOL + HI was higher than the lowest (CTL) by a unit (*p* < 0.0001; 29.3, 28.9, 28.5, and 28.1% for MOL + HI, HI, MOL, and CTL, respectively).

Cellulose content, like the other cell wall carbohydrate contents (NDF, ADF, and hemicellulose), was also lower at 75% irrigation than 100% irrigation by a unit following 5 days of aerobic exposure (Table 7).

Acid detergent lignin (ADL) followed a trend similar to other cell wall carbohydrates, which was lower at 75% irrigation than 100% irrigation by approximately 2 and 4 units for day 0 and 5 days after aerobic exposure, respectively (Table 8). ADL results were significantly different among additives treatments, the lowest ADL was in CTL and highest in MOL + HI (*p* < 0.0018; 8.7, 8.8, 9.4, and 9.5% for CTL, MOL, HI, and MOL + HI, respectively, averaged across irrigation).

Dry matter (DM) loss was lower at 75% irrigation than 100% irrigation for both day 0 and day 5 after aerobic exposure (Table 8). Dry matter loss was significantly different among additives treatments, lowest in CTL, and highest in HI (*p* < 0.0222; 1.8% for all the three additives treatments; CTL, MOL, MOL + HI, and 2.0% for HI, averaged across irrigation).

## 4. Discussion

Tef is known worldwide as a multi-harvest crop with high production capacity and outstanding fodder quality [14,15,18] that could be used as an alternative summer forage [17]. We conducted an experiment to assess the possibility of ensiling tef and the effects of irrigation, genotype, and additives on tef silage. To the best of our knowledge, this is the first report of a study assessing the effects of these factors on tef silage.

On the day of opening the silage bags (day 0), they had a sweet smell, and the colour of the silage was greenish brown with no signs of moulds. This was in contradiction with a study of [42] in guinea grass and cassava tops silages, in which the greenish silage was better than the greenish brown colour which was associated with moulds. The variation in the colour of the silage could probably depend on the ensiling material (plant species) and growth conditions.

It is worth noting that moulds and yeast were not detected on the day of opening or 5 days later, which may indicate the high quality of the ensiling process (Appendix A); however, this phenomenon could be related to the fact that tef grass has remarkable low WSCs content, which did not encourage growth [43]. The moulds and yeast mainly utilise sugars as a substrate [20], and hence could hardly thrive on tef silage. This was also observed during the aerobic exposure of tef silage for five days with production of negligible carbon dioxide (data not shown) without any sign of aerobic spoilage. This is desirable as it minimises losses during the practical feed out phase of the silage.

The pH values at the opening time were lower in 75% irrigation than 100% irrigation regime (Table 3). This could be explained by low moisture content which often leads to better silage fermentation by increasing the counts of LAB, which are essential for rapidly dropping pH, more homogenous silage, selectively inhibiting butyric acid fermentation which is associated with increased pH [44,45,46]. The pH results are similar to the findings of [47], who reported that generally grass silages have a pH of 4.3–4.7. They are also similar to the findings of [46], who reported lower pH values in wilted grass in combination with LAB as an inoculant. The pH of tef silage was slightly greater than that of corn silage, which is usually less than 4 [48,49,50,51], but less than that reported in lucerne [45], as well as less than that reported in ensiled high-moisture corn after 120 days as reported by [52]. The pH of our tef silage was also lower than a mixture of Timothy and Meadow Fescue with additives [53] as well as orchard grass and alfalfa [46]. This could imply that each ensiling crop has different fermentation characteristics, and other factors should be considered in the assessment of the quality of silages. The differences in the pH values were also found between the four genotypes (Appendix A), which could be related to difference in their characteristics [3,13,25,54,55,56]. In addition, the lowest pH values were achieved with the addition of MOL + HI to the tef silage. The lowest pH in MOL + HI could be a result of the synergistic effect between molasses and HI leading to better and rapid fermentation. Assessment of pH after the aerobic stability experiment was even lower for all the additives. This could imply that tef silage has a good buffering capacity and low WSCs, which hamper the multiplication of aerobic microorganisms that could deteriorate the silage. In addition, higher concentrations of acetic acid and VFAs in MOL + HI could act as silage preservatives as described by [57,58,59,60].

In vitro DM digestibility values of tef silage (Table 3) were almost similar to those of whole corn silage as reported by [61]. The IVDMD results were also similar to alfalfa silage [62] and five varieties of sorghum silage [63]. These results imply that tef silage could be used like some of the most common silages. The IVDMD values obtained for the 75% irrigation treatment were greater than those of the 100% irrigation, possibly because of better fermentation with low moisture levels [44,45]. Looking at additives, MOL + HI had the greatest IVDMD, which could be attributed to the synergistic effect between molasses and HI and made nutrients more available for digestion as suggested by [64]. This same trend was observed after the exposure to aerobic conditions for irrigation treatments with IVDMD still higher at 75% than 100% irrigation, though the IVDMD values were lower after 5 days of aerobic exposure. These slightly lower IVDMD values could be attributed to the slight DM loss during the aerobic exposure experiment. However, it was the opposite while looking at the additives; IVDMD was higher in CTL compared to others. This could be as a result of the increased microbial activity that led to more DM degradation in the other three additives than the CTL. The variation in IVDMD among genotypes could be as a result of the differences between the ratio of stems and leaves [3,13,25,54,55,56]. A high stem to leaf ratio could imply less digestibility, while a lower one could imply more digestibility because the leaves are less lignified and hence more digestible than the stems.

The IVDMD values below 50% in two genotypes (RTC-2 and RTC-361) at 100% irrigation following 5 days of aerobic exposure could be attributed to the high indigestible NDF content that was found in these genotypes. This is supported by the marginal differences of digestible NDF between 48 h vs. 240 h of incubation for the varieties as shown in (Table 2).

LAB counts were lower in 75% than 100% irrigation, contradicting with the findings of [44,45] who found more LAB in wilted (lower moisture) silage with lactic acid producing bacteria. This could be attributed to more enzyme activity because most microbial and enzyme activities are controlled by moisture. In addition, the results of lower moisture (wilting) are inconclusive. Although the silage materials had similar DM contents, the materials from 100% had probably more tissue moisture from irrigation than the ones that originated from 75% irrigation. The difference in the genotypes could be attributed to the variation among the tef lines selected for the experiment, as tef is also diverse as reported by [3,13,25,54,55,56]. A higher LAB count in HI treatment relative to the other three treatments is most probably because of the LAB component of HI.

Total VFA concentrations were lower in tef silage than corn silage [65]. The concentrations in MOL + HI were consistently higher in individual VFAs as well as total VFA compared to the other additives. This could imply the synergy between HI and molasses [28]. Certain VFAs, such as butyric and acetic acid, have been implicated in depressing feed intake, although there are some inconsistences [45], hence their lower concentrations might be beneficial. This same trend was maintained during the aerobic stability treatment. However, the VFAs were lower probably because some were lost during the aerobic stability exposure.

Lactic acid was less than 1% in all four treatments, but it was higher in MOL and MOL + HI [28]. This could be attributed to molasses acting as a source of energy for the LAB producing more LA [20] and the synergistic effect between MOL + HI. In comparison between tef silage, LA in other silages from corn, legumes, and grasses was higher (2% or greater) [47,65,66], or almost similar in whole-maize silage (about 1%) and high-moisture corn (0.5–2%) [47]. LA concentration increased during the aerobic stability experiment and was greater at 75% irrigation, which could be attributed to less moisture leading to rapid fermentation activity as well as low WSC [43], which could not support aerobic microorganisms to break down the LA. Phenotypic and genotypic differences could still be the cause of the variation in LA concentrations among the genotypes [3,13,25,54,55,56].

Water-soluble carbohydrates in tef silage was similar to that in high moisture corn silage (<0.10%) as reported by [52], but less than that of corn ensiled with potassium sorbate [65] and those reported on the effect of damaged ears of corn and additives on corn by [66]. WSCs were greater in 75% irrigation. This could be due to lower moisture content leading to better silage fermentation characteristics [67,68,69]. The difference among genotypes could be attributed to the tef diversity reported by [3,13,25,54,55,56]. The lower WSC contents among treatments (0.04 to 0.16 g/100 g DM) and particularly lower in HI and MOL + HI could be attributed to more efficient ensiling where the microorganisms used the sugars as their source of energy. This phenomenon is desirable because the microorganisms’ mass would serve as a rich source of proteins to the animals. The difference among genotypes could still be attributed to variation between of the tef lines. However, the WSC were still lower in HI, MOL, and MOL + HI, which could be attributed to utilisation of the sugars for microbial survival and growth.

Ammonia nitrogen concentration in silages was less in 75% than 100% irrigation. Ammonia nitrogen is a result of proteolysis of amines and other sources of proteins [20]. High N-NH_3_ concentrations could depict less efficient fermentation, and lower ones could depict the conversion of the N-NH_3_ and its derivatives into microbial proteins, which benefits livestock and prevents N wastage [70]. The variation in the N-NH_3_ concentration could still be attributed to diversity [3,13,25,54,55,56], whereas the low concentrations in MOL + HI would imply more efficient utilisation of N-NH_3_ and its derivatives due to the synergy between MOL + HI, as the HI would provide microorganisms for fermentation and molasses would serve as source of available energy. It is also worth noting that there was even more N-NH_3_ at 100% irrigation during the aerobic stability experiment. This increase could be attributed to the likely greater moisture content of the plant material in 100% than 75% irrigation. More moisture content would imply more enzyme and microbial activity leading to more proteolysis; hence, increase in N-NH_3_ concentration and the variation among genotypes could also still be as a result of the diversity in the tef germplasm.

The OM content of tef silages was slightly more in 100% than 75% irrigation, this could be attributed to the faster fermentation where there is more moisture averting losses [47]. The slightly lower OM content of 75% irrigation and lower OM content in HI, MOL and MOL + HI relative to the control could be because the OM was used by the microorganisms. There was a slight variation among genotypes which could still be attributed to differences between lines [3,13,25,54,55,56]. This trend was maintained during the aerobic stability experiment.

Crude protein content in silages was slightly less in 100% than 75% irrigation. This could be because more moisture increases enzyme activity [47], which could have led to more proteolysis than in 75% irrigation. This was demonstrated by the higher N-NH_3_ concentration at 100% vs. 75% irrigation at both day 0 (3.5 vs. 5.0 mgdL^−1^) and day 5 (2.5 vs. 5.6 mgdL^−1^, respectively). A higher N-NH_3_ concentration in silage is undesirable because it is an indicator of protein degradation as suggested by [20]. It is worth noting that although higher in 100% than 75% irrigation, the N-NH_3_ concentration was still below what is usually observed in other silages (8–15%) as reported by [70]. This could imply less protein loss which is desirable in silages. The slight variation among genotypes may still be attributed to tef germplasm diversity [3,13,25,54,55,56]. A similar trend was observed during the aerobic stability exposure. The CP was higher in HI, MOL, and MOL + HI than the control during the aerobic stability experiment. The higher CP could be as a result of microbial proteins accumulated as a result of the additives supplement [64,71].

The cell wall carbohydrates content (NDF, ADF, hemicellulose, and cellulose) in silages and raw material were consistently higher in 100% than 75% irrigation. However, lower fibre content in silage could be attributed to utilisation by microorganisms toward more efficient fermentation [44,45]. Research has shown that the next source of fermentation microorganisms after depletion of WSCs is hemicellulose [20]. Low fibre contents and higher digestible material in silage are desirable because they encourage feed intake and reduce the amount of indigestible fibre. Differences of fibre content among genotypes may still be because of the differences between tef lines. A comparison among silages with additives showed that MOL + HI and HI had less fibre content, which could be indicative of more fibre utilization and efficient fermentation. These results are similar to [64] who reported better fermentation parameters with additives. It is worth noting that after the aerobic stability experiment, the cellulose content was slightly higher in the other three additives treatments relative to the CTL. This could be either because of more DM breakdown leading to concentration of cellulose or more stable fermentation conditions leading to less loss of cellulose.

The ADL content in silages were almost thrice higher in both irrigation treatments than the original ensiling material, but higher in 100% than 75% irrigation. This could be attributed to increased DM loss at more moisture content due to increased microbial and enzyme activity [47] increasing the concentration of ADL. The ADL content was even greater at 100% irrigation after aerobic stability experiment implying more DM loss, whereas it was rather less at 75% irrigation, which could depict better silage fermentation [44,45,67,69] minimizing accelerated DM loss. However, due to a small loss (2–4%) in DM in our experiment, the threefold increase in ADL could not be explained by DM loss, but could be rather a result of artefacts because ADL is hardly degraded or produced during ensiling. Considering the genotypic effect on tef silage, there were variations among genotypes which could be related to the diversity in the tef germplasm, and a further comparison among additives showed that a higher content was seen in the CTL relative to the other treatments. These high ADL contents could be attributed to a more accelerated DM loss in the CTL than other treatments because the additives often mitigate DM losses as reported by [72].

The DM losses were minimal (up to a maximum of 2% and 4% on day 0 and day 5, respectively) This is less than what was observed in other silage crops such as corn, ryegrass and wheat that had between 4 to 11% DM loss [41]. The DM losses in tef silage are also much lower than what was postulated (15 to 30%) by [73]. The low DM losses in tef silage could be attributed the lack of moulds and yeast that are usually implicated in silage deterioration, proper DM content (approximately 35%) during ensiling, and creation of anaerobic conditions [20,22,73] achieved by vacuum sealing. The lower DM loss in tef silage would imply a reduction in wastage and more efficient utilisation of resources. However, the methodology used herein (vacuum-sealed bags) could be a good reason for the minimum DM losses.

A comparison between the two irrigation treatments showed that most of the silage parameters were better at 75% than 100% irrigation. This could be attributed to more rapid vegetative growth and higher yield [74] at 100% irrigation, which would need support structures (lignin). Our analysis of silage raw materials showed that lignin content was higher in 100% irrigation than 75%. Lignification usually reduces the quality of the forage [74,75]. This could be one of the reasons that better silage parameters was achieved at 75% than 100% irrigation. In addition, having better silage parameters at 75% irrigation is desirable, as this would save of the forage production costs (less water usage).

## 5. Conclusions

Our findings show for the first time, at least to the best of our knowledge, that tef can be ensiled despite having low WSCs like other tropical grasses that are more difficult to ensile. This could lean towards the notion that it is not usually the content of the WSC, but rather the desirable DM content, and rapid drop in pH that play a central role in the success of silage fermentation. Our findings also showed that the silage quality was better with additives (HI, MOL and MOL + HI). The results further inferred that silage fermentation parameters were favourable when 75% irrigation was used which could decrease of the costs. Variation between genotypes existed, which is important consideration for tef selection for silage making. However, in vivo feeding experiments are desirable to further complete the assessment of tef silage on production in rations of high yielding dairy cows.

## Figures and Tables

**Table 1 animals-13-00470-t001:** Chemical composition (percentage) of the tef samples before ensiling on DM basis (except DM).

	Irrigation											
	75%				100%					*p*-Value		
Genotype/Item	RTC-2	RTC-119	RTC-361	RTC-400	RTC-2	RTC-119	RTC-361	RTC-400	SEM	Gen	Irr	IrrXGen
DM	30.3 ^ab^	30.8 ^a^	30.4 ^ab^	30.3 ^b^	27.9 ^c^	36.7 ^a^	30.8 ^b^	30.9 ^b^	0.12	<0.0001	<0.0001	<0.0001
OM	90.8 ^b^	90.4 ^b^	92.5 ^a^	93.3 ^a^	88.6 ^b^	90.1 ^a^	91.2 ^a^	87.8 ^b^	0.29	<0.0001	<0.0001	<0.0001
CP	14.0 ^b^	12.2 ^c^	14.8 ^a^	14.4 ^ab^	15.1 ^a^	14.2 ^b^	13.1 ^c^	13.9 ^b^	0.15	<0.0001	0.0074	<0.0001
NDF	66.2 ^a^	66.6 ^a^	62.5 ^b^	63.0 ^b^	68.2 ^b^	70.9 ^a^	67.5 ^b^	69.0 ^ab^	0.45	<0.0001	<0.0001	0.0013
ADF	31.7 ^a^	33.0 ^a^	27.2 ^b^	27.5 ^b^	31.4 ^b^	35.1 ^a^	31.7 ^b^	32.9 ^ab^	0.47	<0.0001	<0.0001	0.0001
Hemicellulose	34.6	33.6	35.3	35.5	36.8	35.8	35.9	36.1	0.62	0.1328	0.0020	0.1536
Cellulose	27.9 ^a^	29.2 ^a^	23.1 ^b^	23.7 ^b^	27.5 ^bc^	29.1 ^a^	26.7 ^c^	28.3 ^ab^	0.40	<0.0001	<0.0001	<0.0001
ADL	3.53	3.68	3.86	3.61	3.83 ^c^	5.76 ^a^	4.95 ^ab^	4.24 ^bc^	0.23	0.0008	<0.0001	0.0027
WSC	0.32	0.42	0.35	0.83	0.77	0.57	0.79	0.35	N/A	N/A	N/A	N/A

WSC results are in g 100^−1^ DM; N/A: Not applicable (because statistical analysis could not be performed). Means in row within the main effect of irrigations with different superscripts differ.

**Table 2 animals-13-00470-t002:** In vitro digestibility (percentage) of the tef samples before ensiling.

	Irrigation											
	75%				100%					*p*-Value		
Genotype/Item	RTC-2	RTC-119	RTC-361	RTC-400	RTC-2	RTC-119	RTC-361	RTC-400	SEM	Gen	Irr	IrrXGen
IVDMD	67.1	57.5	63.1	63.6	58.0	59.4	59.0	65.1	N/A	N/A	N/A	N/A
IVNDFD48 *	54.7 ^a^	44.7 ^b^	44.8 ^b^	46.4 ^b^	42.4 ^c^	45.7 ^b^	45.8 ^b^	53.7 ^a^	0.72	<0.0001	0.0594	<0.0001
IVDMD240 **	73.8	69.1	72.0	70.6	69.2	72.8	72.8	73.7	N/A	N/A	N/A	N/A
IVNDFD240 **	67.1	60.5	59.0	56.9	60.6	65.8	61.1	65.6	3.60	0.4688	0.2253	0.0781

IVDMD: In vitro dry matter digestibility; IVNDFD48 *, in vitro neutral detergent fibre digestibility after 48 h; IVDMD240 **, in vitro dry matter digestibility after 240 h; and IVNDFD240 **, in vitro neutral detergent fibre digestibility after 240 h. N/A: Not applicable (because statistical analysis could not be performed). Means in row within the main effect of irrigations with different superscripts differ.

**Table 3 animals-13-00470-t003:** The effect of irrigation and additives on pH, IVDMD, and LAB of tef silage at day 0 and 5 days after aerobic exposure.

	pH		IVDMD^1^ (%)		LAB^2^ (log_10_ CFU g/DM)
	Day 0	Day 5	Day 0	Day 5	Day 0
Additive	Well-watered (100%)				
CTL	5.00	4.82	56.5	52.9	10.5
HI	4.91	4.78	58.9	50.1	10.4
MOL	4.87	4.83	60.1 *	51.9	10.5
MOL + HI	4.63 ***	4.55 ***	59.4	53.1	10.6
Average	4.85	4.70	58.7	52.0	10.5
	Water-limited (75%)				
CTL	4.21	4.17	62.7	62.4	9.6
HI	4.26	4.17	60.4	56.7	9.7
MOL	4.16	4.08	63.5	58.9	7.3 *
MOL + HI	4.22	4.13	66.1	61.0	8.4
Average	4.20	4.10	63.2	59.7	8.8
Probability of F-Value					
Irrigation	<0.0001	<0.0001	<0.0001	<0.0001	<0.0001
Genotype	<0.0001	<0.0001	<0.0001	<0.0001	<0.0001
Additive	0.0004	0.0040	0.0231	0.0323	0.0109
IrrxGen	<0.0001	<0.0001	<0.0001	<0.0001	<0.0001
IrrxAdd	0.0005	0.0038	0.1062	0.7758	0.0044
GenxAdd	0.0014	0.0017	<0.0001	0.5249	<0.0001
IrrxGenxAdd	<0.0001	0.0015	0.0004	0.0008	0.0002

IVDMD^1^: In vitro *Dry Matter Digestibility*; LAB^2^: Lactic acid bacteria; * and ***: indicate values of additive treatments differing significantly form the control according to Dunnett test at *p <* 0.05 and *p* < 0.001, respectively.

**Table 4 animals-13-00470-t004:** The effect of irrigation and additives on ethanol and volatile fatty acids (VFAs) concentrations (g/100g DM) of tef silage at day 0 and 5 days after aerobic exposure.

	Ethanol		Acetic Acid		Propionic Acid		Butyric Acid		Total	
g/100 g DM										
	Day 0	Day 5	Day 0	Day 5	Day 0	Day 5	Day 0	Day 5	Day 0	Day 5
Additive	Well-watered (100%)									
CTL	0.021	N.D	0.518	0.476	N.D	N.D	0.002	0.024	0.541	0.500
HI	0.036	N.D	1.021 *	0.406	0.021	N.D	N.D	0.022	1.079 *	0.428
MOL	0.025	N.D	0.716	0.667	N.D	N.D	0.018	0.026	0.773	0.693
MOL + HI	0.041	N.D	0.884	0.312	0.012	N.D	N.D	0.136	0.938	0.449
Average	0.031	N.D	0.785	0.465	0.008	N.D	0.005	0.052	0.833	0.518
Additive	Water-limited (75%)									
CTL	0.013	N.D	0.506	0.162	N.D	0.405	0.021	0.024	0.540	0.591
HI	0.001	N.D	0.785	0.372	0.007	N.D	0.008	0.290	0.801	0.662
MOL	0.015	N.D	0.666	0.287	0.058	0.235	0.011	0.025	0.756	0.548
MOL + HI	0.037	N.D	0.850	0.347	0.281	N.D	0.006	0.124	1.173	0.472
Average	0.017	N.D	0.702	0.292	0.087	0.160	0.011	0.116	0.817	0.568
Probability of F-Value										
Irrigation	0.1090	0.5844	0.4397	0.0517	0.1548	0.0006	0.1259	0.1721	0.9005	0.6078
Genotype	0.5888	0.8346	0.2055	0.0704	0.0310	0.0034	0.0839	0.2130	0.9420	0.2312
Additive	0.2772	0.8148	0.0448	0.5600	0.2195	0.0033	0.1079	0.0918	0.0229	0.7290
IrrxGen	0.0109	0.8346	0.0136	0.0328	0.1055	0.0034	0.0019	0.7556	0.0966	0.1647
IrrxAdd	0.5860	0.8148	0.8742	0.2638	0.2426	0.0033	0.1715	0.1026	0.5389	0.5983
GenxAdd	0.9206	0.9646	0.3897	0.5222	0.1085	<0.0001	0.0398	0.4043	0.6255	0.0403
IrrxGenxAdd	0.9366	0.9646	0.4265	0.3212	0.1277	<0.0001	0.0587	0.0680	0.7418	0.0056

N.D: Not detected * indicate values of additive treatments differing significantly form the control according to Dunnett test at *p* < 0.05.

**Table 5 animals-13-00470-t005:** The effect of irrigation and additives on lactic acid, water-soluble carbohydrates, and ammonia nitrogen of tef silage at day 0 and 5 days after aerobic exposure.

	Lactic Acid (g/100g DM)		WSCs^1^ (g/100g DM)		Ammonia Nitrogen (mg/dL)	
	Day 0	Day 5	Day 0	Day 5	Day 0	Day 5
Additive	Well-watered (100%)					
CTL	0.71	0.52	0.10	0.12	6.49	6.49
MI	0.44	0.76	0.05	0.05 *	5.88	5.83
MOL	0.57	1.10	0.06	0.10	6.08	5.90
MOL+MI	0.60	0.70	0.06	0.08	2.32 **	5.19
Average	0.58	0.77	0.07	0.09	5.20	5.85
Water-limited (75%)						
CTL	0.59	1.90	0.22	0.33	3.49	2.24
MI	0.46	1.01	0.03 ***	0.06 ***	3.47	2.99
MOL	0.46	2.89	0.09 ***	0.16 ***	3.25	2.60
MOL+MI	0.41	1.24	0.03 ***	0.11 ***	3.75	2.09
Average	0.48	1.76	0.09	0.16	3.49	2.48
Probability of F-Value						
Irrigation	0.1711	<0.0001	0.0453	<0.0001	0.0029	<0.0001
Genotype	0.0001	0.0035	<0.0001	<0.0001	0.0317	0.0245
Additive	0.1986	0.0058	<0.0001	<0.0001	0.0608	0.6911
IrrxGen	0.0019	0.0023	0.0009	0.0017	0.0334	0.0253
IrrxAdd	0.7380	0.0817	0.0001	<0.0001	0.0174	0.7688
GenxAdd	0.8335	0.6464	<0.0001	<0.0001	0.0082	0.3612
IrrxGenxAdd	0.0005	0.5110	0.0348	0.0043	0.4183	0.4910

WSCs^1^: Water-soluble carbohydrates; *, ** and *** indicate values of additive treatments differing significantly form the control according to Dunnett test at *p* < 0.05, 0.01 and 0.001, respectively.

**Table 6 animals-13-00470-t006:** The effect of irrigation and additives on organic matter, crude protein, and NDF of tef silage at day 0 and 5 days after aerobic exposure.

	Organic Matter (%)		Crude Protein (%)		NDF (%)	
	Day 0	Day 5	Day 0	Day 5	Day 0	Day 5
Additive	Well-watered (100%)					
CTL	89.3	89.7	13.6	13.4	65.9	68.3
HI	89.2	89.6	13.5	13.8	66.5	68.7
MOL	88.7	89.4	13.0	14.1	67.6	69.7
MOL + HI	89.0	89.4	13.8	14.0	66.9	68.8
Average	89.1	89.5	13.5	13.8	66.7	68.9
Water-limited (75%)						
CTL	91.0	91.5	14.4	13.6	62.5	65.3
HI	90.5	90.9 *	14.5	14.6	64.2	67.8
MOL	90.2 *	90.8 *	14.4	14.6	63.4	66.6
MOL + HI	90.4 *	90.9 *	15.0	14.4	61.6	66.7
Average	90.5	91.0	14.6	14.3	62.9	66.6
Probability of F-Value						
Irrigation	<0.0001	<0.0001	<0.0001	0.0190	<0.0001	<0.0001
Genotype	<0.0001	<0.0001	<0.0001	<0.0001	<0.0001	<0.0001
Additive	0.0029	0.0672	0.0315	0.0181	0.1784	0.1378
IrrxGen	<0.0001	<0.0001	<0.0001	0.0004	<0.0001	<0.0001
IrrxAdd	0.7910	0.5623	0.6669	0.6678	0.2621	0.3877
GenxAdd	0.1645	0.4956	0.4151	0.9110	0.0008	0.2874
IrrxGenxAdd	0.1082	0.9935	0.2660	0.6697	0.0069	0.7473

* Indicates values of additive treatments differing significantly form the control according to Dunnett test at *p* < 0.05.

**Table 7 animals-13-00470-t007:** The effect of irrigation and additives on ADF, hemicellulose, and cellulose of tef silage at day 0 and 5 days after aerobic exposure.

	ADF (%)		Hemicellulose (%)		Cellulose (%)	
	Day 0	Day 5	Day 0	Day 5	Day 0	Day 5
Additive	Well-watered (100%)					
CTL	35.8	41.5	30.0	26.7	27.0	29.6
HI	36.4	41.3	30.2	27.4	24.1	31.2
MOL	37.4	42.3	30.3	27.4	24.1	29.9
MOL + HI	37.2	40.0	29.7	28.7	26.1	29.5
Average	36.7	41.3	30.0	27.6	25.3	30.1
Water-limited (75%)						
CTL	34.8	35.8	27.7	29.5	25.2	29.3
HI	34.7	37.5	29.6 *	30.4 *	25.3	28.7
MOL	34.6	37.1	28.8	29.6	25.7	28.9
MOL + HI	33.7	36.8	27.9	29.9	23.9	28.2
Average	34.5	36.8	28.5	29.8	25.0	28.8
Probability of F-Value						
Irrigation	<0.0001	<0.0001	<0.0001	<0.0001	0.5967	0.0119
Genotype	<0.0001	0.0021	0.0021	<0.0001	0.0006	0.0001
Additive	0.7011	0.1519	0.0093	<0.0001	0.3210	0.4970
IrrxGen	<0.0001	<0.0001	0.0503	<0.0001	0.0681	0.0013
IrrxAdd	0.1912	0.1675	0.1126	0.0010	0.0489	0.4613
GenxAdd	0.0037	0.2588	0.0018	0.0040	0.0169	0.2270
IrrxGenxAdd	0.0008	0.6960	0.7292	0.0127	0.0311	0.4103

* Indicates values of additive treatments differing significantly form the control according to Dunnett test at *p* < 0.05.

**Table 8 animals-13-00470-t008:** The effect of irrigation and additives on ADL and DM loss of tef silage at day 0 and 5 days after aerobic exposure.

	ADL^1^ (%)		DM Loss^2^ (%)	
	Day 0	Day 5	Day 0	Day 5
Additive	Well-watered (100%)			
Control	8.1	11.3	2.02	3.59
HI	11.0 ***	9.5 ***	1.94	3.45
MOL	9.4 *	11.8 *	1.84	3.38
MOL + HI	9.5 **	9.8 **	1.71 *	3.39
Average	9.5	10.6	1.88	3.45
Water-limited (75%)				
Control	9.3	6.0	1.51	3.18
HI	8.0 **	8.3 **	2.00 ***	3.33
MOL	8.2 *	7.7 *	1.72 **	3.17
MOL + HI	9.4	8.2	1.87 ***	3.38
Average	8.7	7.6	1.77	3.27
Probability of F-Value				
Irrigation	0.0013	<0.0001	0.0460	0.0296
Genotype	<0.0001	0.3104	0.3460	0.0007
Additive	0.0193	0.4615	0.0222	0.7096
IrrxGen	<0.0001	0.0462	<0.0001	<0.0001
IrrxAdd	<0.0001	0.0168	<0.0001	0.3728
GenxAdd	<0.0001	0.2229	0.7960	0.6214
IrrxGenxAdd	<0.0001	0.1304	0.7716	0.1562

ADL^1^: Acid detergent lignin; DM loss^2^: Dry matter loss; *, **, and *** indicate values of additive treatments differing significantly form the control according to Dunnett test at *p* < 0.05, 0.01 and *p* < 0.001, respectively. Note DM loss was expressed as a percentage of the original sample weight.

## Data Availability

Supporting data of this research will be available on request from the corresponding author.

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
