# Peer review of "The Effects of Irrigation, Genotype and Additives on Tef Silage Making"

_animals, 2023, doi:10.3390/ani13030470_

Round 1
Reviewer 1 Report
The paper submitted for review "”Tef – a potential new crop for high quality silage” is consistent with the profile of the journal "Animals”.
The research is concerned with evaluating the possibility of using Tef (Eragrotis tef) for silage production in ruminant nutrition.
The study concluded that they are a potential source for silage production. Which is beneficial especially since this plant shows high productivity under high radiation and temperature conditions.
The paper is interesting and clearly written.
In my opinion, it is suitable for publication in its present form.
Author Response
Thank You so much for Your comments.
Reviewer 2 Report
Dear authors,
I have reviewed the manuscript entitle “Tef – a potential new crop for high quality silage”. I agree with the use of local feedstuffs to feed livestock and tef is one of them. Regarding your research, I notice a lot of work and a lot of results. Below you will find my comments:
Title:
The title sounds more like a review article than a research one. I highly recommend changing the title so that it better reflects the research you performed and according with the aim of the study.
Simply summary
Line 15: “m” is missing in molasses+heterofermentative.
Introduction
In general is complete and well written.
Materials and methods
Plant materials and growth conditions
Line 90-93: I recommend you to include a figure (as supplementary file) about the design of irrigation regimes.
Silage making
Line 109-110: Delete the sentence. This is part of results shown in Table 1.
Line 111: particle size
Chemical analyses of tef silage
Line 159-161: IVDMD technique (Tilley and Terry, 1963) requires ruminal fluid. Indicate how you collect the ruminal fluid, type of ruminant, diet, etc.
Indicate measurement time.
Approval from the appropriate ethics committee is mandatory.
Results
Table 1. This table is too large and not clear. I suggest two tables: 1. Chemical composition and 2. Digestibility. Is it correct IVDMD240? What is the difference between IVDMD with IVDMD240h?
In general, I recommend you to improve tables.
Discussion
Well written.
Line 524: Include the Institutional Review Board Statement.
Author Response
|
Line |
Reviewer comments |
Response |
|
Title |
The title sounds more like a review article than a research one. I highly recommend changing the title so that it better reflects the research you performed and according with the aim of the study. |
The title was changed to “The effects of irrigation, genotype, and additives on Tef silage”.
|
|
15 |
“m” is missing in molasses + heterofermentative |
the “m” was added in molasses |
|
90-93 |
I recommend you include a figure (as supplementary file) about the design of irrigation regimes
|
A Figure (S1) was included in supplementary file |
|
109-110 |
Delete the sentence. This is part of results shown in Table 1 |
This was deleted.
|
|
111 |
Particle size |
6 cm was not the particle size (for chemical analysis), it was the size of the ensiling material. |
|
159-161 |
IVDMD technique (Tilley and Terry, 1963) requires ruminal fluid. Indicate how you collect the ruminal fluid, type of ruminant, diet, etc. Indicate measurement time. Approval from the appropriate ethics committee is mandatory. |
This was indicated in details.
This was indicated. All experiments were done after approval by the IACUC of the Hebrew University of Jerusalem (AG-14102). This was indicted in the Ethical declaration section and M&M. |
|
Table 1 |
This table is too large and not clear. I suggest two tables: 1. Chemical composition and 2. Digestibility. Is it correct IVDMD240? What is the difference between IVDMD with IVDMD240h? |
This was done as suggested. Now there is Table 1 and Table 2. IVDMD240 is correct (we added as well explanation in materials and methods section). This was done by incubation the material for 240h in the Daisy system while IVDMD was done according the protocol pf Tilly and Terry (as described in M&M section). |
|
In general |
I recommend you to improve tables. |
We improved Table 1 as suggested. However, the rest we believe they are clear for the reader. Nonetheless, we will improve them according to specific suggestions. |
|
Discussion |
Well written. |
Thank you |
|
524 |
Include the Institutional Review Board Statement. |
We don’t have Institutional Review Board |
Reviewer 3 Report
This results of this research study compared the difference point between Day 0 and Day 5. In the case of explanation of LAB fermentation part, I wondering that Day 5 might be as the Death stage of LAB growth curve. So, it would be impact the results for LAB or could be as a false positive error. Could you please give some idea to support the data on Day 5? Please showed the strong evidence why the Day 5 is considered to use in this research study. It would be clearly explanation and fit with this research data.
Author Response
Thank you for your comments. We did not count LAB after 5 days of aerobic conditions exposure. Day 5 refers to aerobic stability experiment (5 days of exposing the silage to oxygen) as detailed in (Ashbell et al., 1991). The 5 days exposure is accepted as the standard period for examining the silage aerobic stability and growth of aerobic microflora (moulds, yeast, and aerobic bacteria).
Round 2
Reviewer 2 Report
Dear authors,
I hope my suggestions have helped to improve your manuscript.
Good luck!
Author Response
Dear reviewer,
Thank you very much for your helpfull comments and suggestions. We did our best to address them.